# The Low-Waste Grafting Copolymerization Modification of Chitosan Is a Promising Approach to Obtaining Materials for Food Applications

**DOI:** 10.3390/polym16111596

**Published:** 2024-06-04

**Authors:** Maria S. Lavlinskaya, Andrey V. Sorokin, Anastasia A. Mikhaylova, Egor I. Kuznetsov, Diana R. Baidamshina, Igor A. Saranov, Margaryta V. Grechkina, Marina G. Holyavka, Yuriy F. Zuev, Ayrat R. Kayumov, Valeriy G. Artyukhov

**Affiliations:** 1Biophysics and Biotechnology Department, Voronezh State University, 1 Universitetskaya Square, 394018 Voronezh, Russia; andrew.v.sorokin@gmail.com (A.V.S.); holyavka@rambler.ru (M.G.H.); artyukhov@bio.vsu.ru (V.G.A.); 2Polymer Science and Colloid Chemistry Department, Voronezh State University, 1 Universitetskaya Square, 394018 Voronezh, Russia; minas36@yandex.ru (A.A.M.); kuznetsoffegorr@gmail.com (E.I.K.); 3Institute of Fundamental Medicine and Biology, Kazan (Volga Region) Federal University, 18 Kremlevskaya Street, 420008 Kazan, Russia; dianabaidamshina@yandex.ru (D.R.B.); kairatr@yandex.ru (A.R.K.); 4Research Core Center “Testing Center”, Voronezh State University of Engineering Technologies, 19 Revolutsii Avenue, 394036 Voronezh, Russia; mr.saranov@mail.ru; 5Research Core Center, Voronezh State University, 1 Universitetskaya Square, 394018 Voronezh, Russia; grechkina_m@mail.ru; 6Physics Department, Sevastopol State University, 33 Studencheskaya Street, 299053 Sevastopol, Russia; 7Kazan Institute of Biochemistry and Biophysics, FRC Kazan Scientific Center of the RAS, 2/31 Lobachevsky Street, 420111 Kazan, Russia; yufzuev@mail.ru

**Keywords:** chitosan, graft polymerization, poly(*N*-vinylpyrrolidone), antimicrobial activity

## Abstract

Chitosan takes second place of the most abundant polysaccharides naturally produced by living organisms. Due to its abundance and unique properties, such as its polycationic nature, ability to form strong elastic porous films, and antibacterial potential, it is widely used in the food industry and biomedicine. However, its low solubility in both water and organic solvents makes its application difficult. We have developed an environmentally friendly method for producing water-soluble graft copolymers of chitosan and poly (*N*-vinylpyrrolidone) with high grafting efficiency and a low yield of by-products. By using AFM, SEM, TGA, DSC, and XRD, it has been demonstrated that the products obtained have changed properties compared to the initial chitosan. They possess a smoother surface and lower thermal stability but are sufficient for practical use. The resulting copolymers have a higher viscosity than the original chitosan, making them a promising thickener and stabilizer for food gels. Moreover, the copolymers exhibit an antibacterial effect, suggesting their potential use as a component in smart food packaging.

## 1. Introduction

Chitosan (Cht) is a product of the partial deacetylation of the natural polymer chitin, composed of *N*-acetyl-D-glucosamine links. It is one of the most common natural polysaccharides found in crustacean shells and fungal cell walls. The large volume of research and practical interest in chitin and chitosan are driven by their unique structures, which, due to the presence of both hydroxyl and amine groups, lead to a variety of functional features. Moreover, Cht is characterized as a biocompatible, low-toxic, biodegradable polymer possessing a high sorption capacity for a multitude of compounds [1,2,3,4,5].

Along with its diverse biological activity, these properties determine the widespread use of Cht in the food industry: the antioxidant and antimicrobial activities of chitosan make it possible to use it for smart packaging development, increasing the shelf life of food [6], while the excellent emulsifying properties of Cht allow for the replacement of common surfactants in food technologies [7]. Additionally, the high viscosity of chitosan solutions defines their application as either a thickener or gel-forming agent [8,9]. It has been shown that the chitosan coating of fresh-cut or whole kiwifruit preserves consumer properties by decreasing certain parameters such as respiration activity, weight loss, and firmness and delaying ripening [10]. Moreover, Cht coatings are promising in meat product storage due to their reduction of lipid oxidation and discoloration [11]. However, the use of chitosan in food packaging is limited due to its low solubility at a neutral pH [12]. Its low mechanical resistance and uncontrolled biodegradability also limit the formation of films with desirable properties. To overcome these disadvantages, various chitosan modification techniques have been proposed.

Graft copolymerization is one of the most widespread polymer modification approaches used to prepare hybrid polymers with enhanced properties from natural polysaccharides. In this technique, polymer side chains grow from the polysaccharide backbone (*grafting from*) or from pre-obtained polymer grafts to the main backbone (*grafting to*) [13]. By varying the properties of the side chains, the parameters of the macromolecules obtained can be tuned; for example, the grafting of water-soluble side chains enhances chitosan’s solubility in aqueous media. A promising candidate capable of increasing chitosan’s solubility and viscoelastic properties is poly(*N*-vinylpyrrolidone) (PVP). PVP is a low-toxic, film-forming, biocompatible polymer widely used in pharmacy as the base for drug formulations or as a blood substitute [14,15]. It also has surfactant properties [16]. On the one hand, the combining of chitosan and poly(*N*-vinylpyrrolidone) links in one macromolecule allows us to obtain a soluble, neutral pH, elastic, film-forming compound with antibacterial properties. On the other hand, the grafting of PVP onto chitosan poses some difficulties. Firstly, the obtaining of a uniform side chain distribution requires the polymerization to occur in solution. A water solution is cheaper and aligns with the green chemistry trend, as well as the safety recommendations for the food industry. Moreover, in the case of chitosan, this solution must have a pH < 6.5. It is well known that *N*-vinylpyrrolidone is poorly polymerized in acidic mediums due to side processes (Figure 1) [17]. These side processes lead to an increase in grafting efficiency and monomer conversion, reducing the prospects for the practical application of the resulting products.

To initiate polymerization, various methods are employed, such as irradiation (γ, UV, electron beam, etc.) or substance initiation. In the case of irradiation initiation, numerous side processes, such as carbohydrate destruction, may occur [18,19]. Substance initiation involves the use of compounds capable of forming radicals on the carbohydrate backbone. The commonly employed compounds like Ce(IV), Fe(II), and other heavy metal salts are undesirable in food production due to their toxicity to humans. Potassium persulfate (PPS), a water-soluble and low-toxic compound, is preferable, but its high oxidation potential may lead to the destruction of Cht macromolecules. Therefore, the careful selection of polymerization conditions for chitosan and *N*-vinylpyrrolidone is necessary to obtain promising materials for food technology.

Here, we report optimized synthesis conditions for graft copolymers of chitosan and poly(*N*-vinylpyrrolidone) (Cht-*g*-PVP) as a form of low-waste graft copolymer production and demonstrate their properties, with a focus on their potential applications in the food industry.

## 2. Materials and Methods

### 2.1. Materials

Chitosan (Bioprogress, Losino, Moscow Region, Russia) with a molecular weight of 350 kDa and a deacetylation degree of 0.85 was chosen for modification. Before use, it was dried to a constant weight at 60 °C. *N*-vinylpyrrolidone (VP; >99%, Sigma, Saint Louis, MO, USA) with a boiling point of 79–81 °C at 5 mm Hg was purified by vacuum distillation. Potassium persulfate (PPS) and sodium metabisulfite (MBS) (both >98%, Vekton, Saint Petersburg, Russia) were recrystallized from water.

### 2.2. Synthesis of Graft Copolymers of Chitosan and N-Vinylpyrrolidone

For polymerization, 0.5 g of chitosan and 50 mL of a 2% (*w*/*w*) acetic acid solution were placed in a Schlenk flask equipped with a magnetic stirrer and an argon purge line. The calculated amount of *N*-vinylpyrrolidone (the molar ratios of Cht/VP were 1/3, 1/5, and 1/10) was added to the flask after the chitosan was completely dissolved. Then, the obtained mixture was degassed by three freeze–pump–thaw cycles. The initiator mixture, consisting of the calculated potassium persulfate (PPS) and sodium metabisulfite (MBS), was added against the argon flow. After polymerization was completed, the mixture was poured into acetone, and the precipitate was isolated by centrifugation. The crude copolymer, containing homopolymer, was dried under vacuum, weighed, and purified in a Soxhlet apparatus with ethanol to extract the homopolymer and other impurities. The final product was vacuum-dried to a constant weight.

### 2.3. Characterization of the Cht-g-PVP Copolymers

#### 2.3.1. FTIR-ATR

FTIR-ATR spectroscopy was used to confirm the copolymers’ formation. The FTIR-ATR spectra of powder samples with a resolution of 4.0 cm^−1^, in the frequency range of 800–4000 cm^−1^, 64 scans per one spectrum, were recorded using a Bruker Vertex-70 spectrometer (Brucker Corporation, Billerica, MA, USA) equipped with an attenuated total reflectance attachment with a single reflection diamond working element.

#### 2.3.2. ^1^H NMR Spectroscopy

The ^1^H NMR spectra of the chitosan and graft copolymers were recorded using a Bruker AVANCE 400 MHz spectrometer (Brucker Corporation, Billerica, MA). Samples were dissolved in a 2% (*w*/*w*) CD_3_COOD solution in D_2_O with a polymer concentration of 3 mg × mL^−1^. TMS was used as an internal standard.

#### 2.3.3. SEM

The surface morphology of the polymers was studied by scanning electron microscopy using a JSM-6510LV scanning electron microscope (Jeol Ltd., Tokyo, Japan) in SEI mode. Prior to photography, a 10 nm gold layer was sprayed onto the samples.

#### 2.3.4. AFM

The topography and roughness of the polymer surface were examined by AFM using a Solver P47 Pro (NT-MDT Spectrum Instruments, Moscow, Russia) scanning probe microscope equipped with an NSG03 cantilever device (NT-MDT Spectrum Instruments, Moscow, Russia). Data on surface pores’ presence and depth were obtained by analyzing a section of the 3D maps of the sample surfaces.

#### 2.3.5. Polymer Thermal Research

The thermal stability of the polymers was studied using the simultaneous thermal analyzer STA 449F3 Jupiter (Netzsch, Selb, Bayern, Germany), via TGA and DSC measurements. The samples were placed in aluminum crucibles in a nitrogen atmosphere with a temperature range of 28–600 °C and a heating rate of 10 °C × min^−1^. Indium was used as the calibration standard.

#### 2.3.6. Characterization of Grafted PVP Chains

To determine the molecular weight of the grafted PVP chains, the synthesized Cht-*g*-PVP copolymers were subjected to oxidative degradation [20] followed by GPC analysis performed on an Agilent 1200 Series instrument (Agilent Technologies, Santa Clara, CA, USA) equipped with an isocratic pump, a refractometric detector, and a PLmixC column. A solution of isolated PVP chains was prepared in *N*-methylpyrrolidone with a concentration of 1 mg × mL^−1^. The eluent used was 0.03 M LiCl solution in *N*-methylpyrrolidone, the chromatography temperature was 50 °C, the flow rate was 0.5 mL × min^−1^, and the volume of the injected sample was 20 µL. Calibration was carried out using narrowly dispersed polystyrene standards.

Monomer conversion, *C*, was calculated using the following equation:(1)$$C=\frac{m_{1}-m_{Cht}}{m_{VP}}\times 100$$
where *m*_1_, *m_Cht_*, and *m_VP_* are masses of the non-purified in a Soxhlet apparatus of the graft copolymer, chitosan, and *N*-vinylpyrrolidone, *g*, respectively.

Homopolymer formation, *M*, was evaluated as follows:(2)$$M=m_{1}-m_{2},$$
where *m*_1_ and *m*_2_ represent the mass of the non-purified sample in a Soxhlet apparatus and the mass of the purified graft copolymer, respectively.

The grafting efficiency, *GE*, was calculated using the following equation:(3)$$GE=\frac{m_{Cht-g-PVP}-m_{Cht}}{m_{VP}}\times 100,$$
where *m_Cht-g-PVP_*, *m_Cht_*, and *m_VP_* are the mass of the graft copolymer obtained, and the chitosan and *N*-vinylpyrrolidone used in the polymerization, g, respectively.

The PVP weight content in the copolymer, denoted as *PVP*, %, was calculated as follows:(4)$$PVP,\;\%=\frac{m_{1}}{m_{2}}\times 100$$
where *m*_1_ and *m*_2_ are the masses of isolated PVP chains and copolymer used in the destruction, g, respectively.

The frequency of grafting, *FG*, is expressed as the number of grafted polymer chains per anhydrous glucosamine unit, *AGU*, in the backbone polymer and is obtained from the relationship [21]
(5)$$FG=\frac{PVP,\;\%}{M_{PVP}}\times \frac{M_{AGU}}{Cht,\;\%},$$
where *PVP*, % is the percentage of grafted PVP in Cht-*g*-PVP; *M_PVP_* is the molecular weight of PVP; *M_AGU_* is the average weight of an anhydrous glucose unit of Cht; and *Cht*, % is the percent of Cht in Cht-*g*-PVP.

#### 2.3.7. XRD

XRD patterns were obtained with an Empyrean B.V. diffractometer (Malvern Panalytical B.V. Ltd., Malvern, UK), equipped with a Cu-Kα radiation source (λ = 1.54 nm, 45 kV, and 35 mA), in the scattering angle range of 2θ from 5° to 50°, with a resolution of 0.1° and a scanning speed of 0.2° × min^–1^.

#### 2.3.8. Viscosity Measurement

The viscosity measurements of 0.4% (*w*/*w*) solutions in 2% (*w*/*w*) acetic acid, for chitosan, or in deionized water, for graft copolymers, were performed at 20 °C using an AND SV-1A vibro viscosimeter (AND Company, Tokyo, Japan) in a thermostated cuvette. Each sample was measured in triplicate, and the value was presented as the average ± standard deviation.

### 2.4. Antibacterial and Toxicity Assays

The antibacterial properties of compounds have been studied by determining their minimum inhibitory concentrations (MICs) using a broth microdilution assay in 96-well plates, according to the EUCAST rules for antimicrobial susceptibility testing [22], in full Mueller-Hinton broth (MH) against *Staphylococcus aureus* ATCC 29213 (MSSA) and *Escherichia coli* ATCC 25922. The concentrations of compounds to be tested ranged from 1 to 2048 µg × mL^−1^. The MIC was determined as the lowest concentration of antibiotic at which no visible bacterial growth could be observed after 24 h of incubation.

The cytotoxicity of compounds was determined using the MTT assay (MTT, 3-(4,5-dimethylthiazol-2-yl)-2,5-diphenyltetrazolium bromide, Sigma-Aldrich, Saint Louis, MO, USA) [23] on bovine embryonic lung epithelial cells, LEK (from the Russian Collection of Cell Cultures of Vertebrates (CCCV)). Cells were grown in DMEM—Dulbecco’s Modified Eagle’s Medium (PanEco, Saint-Petersburg, Russia)—supplemented with 10% FBS (Biosera, Cholet, France), 2 mM L-glutamine, 100 µg × mL^−1^ penicillin, and 100 µg × mL^−1^ streptomycin, in 96-well plates with a density of 3000 cells per well, and cultured at 37 °C and 5% CO_2_. After 24 h, compounds were added in concentrations ranging from 1 to 8096 µg × mL^−1^, and cultivation was continued for the next 48 h. Residual metabolic activity was assessed by the MTT assay. The concentrations of polymers leading to a half-reduction of the cells’ viability (CC_50_ values) were calculated using GraphPad Prism 6.0 software (GraphPad Software) with the four-parameter model and agonist concentrations in log-scale.

## 3. Results and Discussion

### 3.1. Optimization of the Cht-g-PVP Synthesis Conditions

The development of functional bio-based materials holds promise for enhancing their application in diverse industries. Moreover, these processes often align with green trends by excluding toxic components such as organic solvents and heavy metal ions. However, combining natural components, such as carbohydrate polymers, which are often (limitedly) soluble only in aqueous media, with synthetic components, such as vinyl monomers, which react preferably in non-aqueous environments, requires special conditions to ensure the high accessibility of the carbohydrate and monomer functionality. In this work, we aim to optimize the synthesis conditions of graft copolymers of chitosan and poly(*N*-vinylpyrrolidone) without using heavy metal ions or irradiation and minimize the destructive effects of acidic media and PPS oxidation on *N*-vinylpyrrolidone and chitosan, respectively.

Data analysis reveals that, due to different side processes, the conversion of *N*-vinylpyrrolidone in grafting-to-chitosan reactions, e.g., its effective conversion to a target product, occurring in acidic water solutions is not high, reaching values less than 15%, followed by a high homopolymer yield of up to 42% [24,25,26]. Such a high by-product yield renders the process ineffective and also increases time and financial costs when isolating the target product. Moreover, the use of organic solvents is required to purify the copolymer, which does not align with modern green trends. Therefore, this work aims to optimize the synthesis of Cht-*g*-PVP copolymers while maximizing their compliance with green trends.

The synthesis of Cht-*g*-PVP copolymers was carried out via aqueous solution radical polymerization with substance initiation. A mixture of potassium persulfate (PPS) and sodium metabisulfite (MBS) served as the initiator. The choice of initiator was determined by various factors. The use of common initiating systems based on metal ions (Ce^4+^, Fe^2+^–H_2_O_2_, etc.) is undesirable due to the challenge of purifying the final product of residues of the initiators, which are salts of heavy metals that can have negative effects on the human body. This problem can be circumvented by employing material initiators such as potassium persulfate.

PPS finds applications in both the food and cosmetology industries and is classified by the US FDA as generally recognized as safe (GRAS) for use as a component of coatings on fresh citrus. It is known that the use of PPS as a polymerization initiator can lead to the degradation of polysaccharides and requires high temperatures, resulting in decreased yields of the target product [27]. However, the use of an initiating mixture containing sodium metabisulfite reduces the temperature of PPS’s decomposition and also protects the polysaccharide from degradation, thereby increasing the efficiency of graft polymerization [28]. Sodium metabisulfite is also classified by the US FDA as GRAS and finds wide application in the food and pharmaceutical industries. Therefore, the use of such an initiating system is justified when modifying chitosan for use in the food industry.

During our research, we discovered that, with an excess of PPS, i.e., *n* (MBS) < *n* (PPS) (mol/mol), the grafting efficiency decreases both below and above the temperature of PPS decomposition, which is approximately 50–55 °C (Table 1). In the former case, this decrease may result from an insufficient concentration of free radicals due to an only partial decomposition of PPS, while, in the latter case, it is attributed to chitosan degradation at higher temperatures. A decrease in grafting efficiency is also noted when using an initiating system in which the metabisulfite exceeds the persulfate (*n* (MBS) > *n* (PPS)). This could be explained by the accumulation of an excess amount of sulfate ion radicals, leading to the termination of growing grafted chains [29].

As a result of our research, we determined that the optimal composition of the initiating system is an almost equal molar ratio of the components, i.e., *n* (MBS) ≈ *n* (PPS). In this scenario, the highest grafting efficiency and effective monomer conversion, reaching 19.8 and 34.5%, respectively, are observed.

The concentration of the initiator greatly influences the polymerization rate and the characteristics of the resulting graft copolymers. At low concentrations, the polymerization reaction slows down significantly, resulting in decreased monomer conversion. Conversely, synthesis with a high concentration of initiator increases the likelihood of side processes, such as polysaccharide and reaction product degradation, and leads to a higher proportion of homopolymer. The results of selecting optimal conditions for the synthesis of Cht-*g*-PVP copolymers are presented in Table 1.

For the molar component ratio *n* (Cht)/*n* (VP) = 1/5, the kinetics of polymerization were investigated (Figure 1). It was observed that both the kinetic and grafting efficiency profiles exhibit a clear extremum. The decrease in graft copolymer yield after this maximum is apparently due to chitosan degradation prevailing over the growth of polymer chains as the concentration of free monomer in the reaction mixture decreases. Consequently, polymerization should be halted upon reaching this maximum. An acidic reaction medium adversely affects the polymerization of *N*-vinylpyrrolidone [30], but, as the grafting efficiency profile indicates, if the reaction is stopped at the extremum point, a significant proportion of monomer is grafted onto the polysaccharide chain.

The characteristics of the Cht-*g*-PVP copolymers obtained and PVP chains grafted under optimal conditions are presented in Table 2. As can be seen from the data obtained, the monomer concentration affects grafting efficiency and frequency as well as *N*-vinylpyrrolidone conversion. With an increase in the VP ratio in the polymerization mixture, the molecular weight of the grafted PVP chains, the PVP fraction in the copolymers, and the frequency of grafting increase. At the same time, an increase in monomer concentration leads to a decrease in conversion and grafting efficiency. This is probably due to the influence of the viscosity medium, which increases with VP concentration growth. Additionally, the products obtained are water-soluble, unlike non-modified chitosan.

Therefore, upon conducting this research, we can identify the optimal conditions that yield a high *N*-vinylpyrrolidone conversion (up to 34.5%) and grafting efficiency (up to 19.8%) in the grafting-to-chitosan reaction occurring in a green reaction medium without heavy metal ions or organic solvents. Moreover, the suggested reaction process is characterized by low waste and by-product yields.

### 3.2. Characterization of Cht-g-PVP Copolymers

To confirm the structure of the resulting copolymers, the FTIR-ATR and ^1^H NMR spectroscopy methods were used. Figure 2 shows the FTIR-ATR spectra of unmodified chitosan and the Cht-*g*-PVP copolymers obtained. The FTIR-ATR spectrum of chitosan contains the following characteristic absorption bands: 2874 cm^−1^ for the asymmetric and symmetric stretching of C-H bonds, a position typical for carbohydrate polymers; residual *N*-acetyl groups appear around 1645 cm^−1^ (C=O stretching, amide I); the amide II band (N-H bending and C-N stretching) is observed at 1565 cm^−1^; the bending of the CH_2_ and CH_3_ groups is at 1420 and 1375 cm^−1^, respectively; the narrow band at 1150 cm^−1^ corresponds to the C-O-C glycosidic bond absorption; and the components at 1060 and 1025 cm^−1^ are attributed to the stretching of the C-O bond in the pyranose rings [31]. The FTIR-ATR spectra of the Cht-*g*-PVP copolymers obtained contain the absorption bands described above. Additionally, new low-intensity bands appear in the spectra of polymers at 1458–1463 cm^−1^, related to the bending of the CH_2_ groups of the chain and the lactam rings of poly(*N*-vinylpyrrolidone) [32]. These changes confirm the formation of Cht-*g*-PVP copolymers.

The ^1^H NMR spectra also confirm the formation of Cht-*g*-PVP copolymers. The NMR spectrum of chitosan solution contains a high-intensity complex signal in the region of 2 ppm. This signal arises from overlapping signals of the CD_2_H fragment in CD_3_COOD, formed as a result of proton exchange between the residual water and the CH_3_ group of the chitosan acetyl residue. The correlation between the proton signals of the chitosan glycosidic rings is presented in Figure 3a [33]. The signals of these protons are broad bands characterized by chemical shifts of 4.86 (H1), 3.91–3.71 (H3–H6), and 3.17 ppm (H2). The NMR spectra of Cht-*g*-PVP copolymers contain both the aforementioned and new signals. The signals of PVP backbone protons are located in the regions of 1.20–1.53 and 3.40–3.56 ppm, while the lactam ring protons are reflected in the regions 2.11–2.14, 2.2–2.5, and 3.41–3.43 ppm [34] (Figure 3b,c). Interestingly, the signals of the grafted poly(*N*-vinylpyrrolidone) chains are narrower compared to those of chitosan. This difference in signal width is due to a significant difference in the molecular weights of the chitosan and grafted PVP chains (Table 2), making it difficult to quantify the copolymers obtained using NMR data.

To investigate the surface morphology of the Cht-*g*-PVP copolymer obtained, SEM and AFM methods were used. As seen from the presented data, non-modified chitosan exhibits a scaly surface typical of some polysaccharides (Figure 4a). Graft copolymerization “smooths” the copolymers’ surface morphology, providing additional confirmation of the grafting of synthetic polymer chains onto the chitosan matrix (Figure 4b). However, areas with surface morphologies belonging to the original polysaccharide are still observed in the SEM images. The largest number of such areas is observed for the Cht-*g*-PVP-1 copolymer with the highest PVP content (Figure 4c,d).

The SEM results, indicating a smoothing of the chitosan surface after its modification by grafted PVP chains, are also confirmed by the AFM data. Figure 5a shows the surface topography and cross-section of the original chitosan over an area of 10 × 10 μm. As seen from the data presented, the surface of chitosan exhibits a rugged relief with various irregularities, protrusions, and depressions, forming a scaly surface characteristic of polysaccharides. Additionally, a significant number of pores varying in diameter, shape, and depth, up to 40 nm, are observed. The maximum height difference is 94.3 nm, and the average roughness is 8.79 nm.

Figure 5b illustrates the surface topography and cross-section of the Cht-*g*-PVP-3 graft copolymer with the highest PVP content over an area of 10 × 10 µm. Pores with a depth of about 40 nm are still observed on the surface, albeit in much smaller quantities. Generally, the cross-section profile is less prominent compared to the original polysaccharide, and its scaly structure is much less pronounced. The surface appears smoothed, with height differences of no more than 27 nm for non-porous regions; the average roughness of the sample is 6.3 nm.

Stability, including thermal stability, is one of the main parameters that determine the possibility of using a material as a component of a food product subjected to heat treatments. Figure 6a shows TGA profiles reflecting the polymers’ weight loss depending on temperature. The first stage of chitosan mass loss (~9%) is observed in the temperature range of 28–100 °C and is due to the loss of the water associated with the polysaccharide. Afterward, the curve reaches a plateau, maintaining a constant sample weight up to 248 °C. Subsequently, in an inert medium, the non-oxidative thermal destruction of the polysaccharide occurs, which is observed in the temperature range of 248–600 °C, resulting in the deacetylation of chitosan, evaporation, and the elimination of volatile decomposition products [35]. The degradation of chitosan begins with the formation of unsaturated structures by amino groups [36]. During polysaccharide pyrolysis, the random cleavage of glycosidic bonds occurs, followed by decomposition with the formation of acetic, butyric, and a number of lower fatty acids, which contain two, three, or six carbon atoms [37].

The moisture loss in the temperature range of 29–100 °C for the Cht-*g*-PVP-3 copolymer containing the largest content PVP is characterized by a higher rate and accounts for about 12% of its initial mass. Thus, the moisture content in these samples is higher than in the original chitosan, emphasizing the hydrophilic nature of the polysaccharide’s modification. The TGA profile of Cht-*g*-PVP-1 is more similar to the one obtained for chitosan, which is due to its low content of grafted PVP chains. Additionally, the thermograms of graft copolymers have a flatter shape, and their residual masses are lower than those of the unmodified chitosan. This difference is quite noticeable in the region of 400–600 °C, where the thermal decomposition of PVP occurs [38]. Thus, the grafting of hydrophilic poly(*N*-vinylpyrrolidone) chains reduces the thermal stability of chitosan.

The decrease in the thermal stability of Cht-*g*-PVP copolymers is also confirmed by the DSC results. The endothermic peak of chitosan melting is observed at 264 °C, whereas, for graft copolymers, it occurs in the region of 231.5–239 °C (Figure 6b). A strong exothermic peak in the DSC profile of chitosan at 294 °C corresponds to its thermal decomposition. For graft copolymers, these signals are in the range of 234–242 °C. It is also worth noting that, for the copolymer with the highest PVP content, Cht-*g*-PVP-3, exothermic peaks of the decomposition of PVP links are observed at 404 and 411 °C, respectively [38], while, for the copolymer with the lowest PVP content, Cht-*g*-PVP-1, the signal is less pronounced, which is due to its lower content of grafted chains.

To elucidate the results obtained from these thermal studies, XRD patterns of chitosan and the Cht-*g*-PVP copolymers were analyzed. The XRD pattern of chitosan (Figure 7) exhibits characteristics typical of semicrystalline polymers, consisting of repeating reflections with decreasing signal intensity. The prominent maximum at 20.3° corresponds to (200), which is characteristic of chitosan [39]. In contrast, for Cht-*g*-PVP copolymers, an absence of pronounced reflections is observed; instead, the signals form a blurred halo, typical of amorphous materials. This indicates a disruption of the crystalline structure of chitosan, which is formed by intra- and intermolecular hydrogen bonds. The grafting of amorphous PVP chains occurs through the amino and hydroxyl groups of chitosan, which are directly involved in the formation of the polysaccharide’s crystalline structure. The introduction of amorphous chains disrupts the distribution of its hydrogen bonds, leading to the destruction of its internal structure and a decrease in its melting and decomposition temperatures.

So, the grafting of PVP chains onto the chitosan matrix alters its inner structure and surface morphology, disrupting its hydrogen bond network and smoothing its scaly surface.

### 3.3. Prospects for the Application of Cht-g-PVP Copolymers in the Food Industry

Considering the main fields of application for chitosan in the food industry, it is evident that promising avenues include its use as a gelation agent or thickener and as a component in food packaging. It is well-known that, depending on its concentration, chitosan forms highly viscous solutions or gels, which can serve as a base for various food products such as jelly, marmalade, etc. On the other hand, chitosan can also form elastic films suitable for food packaging, providing protection for fruits, meat, and other products against microbial contamination, rotting, and browning. In this context, we assess how the grafting of PVP chains affects chitosan’s viscosity and its antimicrobial properties.

Figure 8 presents the influence of PVP chain grafting on the viscosity of 0.4% *w*/*w* polymer solutions. As seen from the data obtained, the presence of PVP chains affects chitosan’s viscosity differently. In the case of the Cht-*g*-PVP-1 copolymer, the viscosity is partially equal to that of the original chitosan, whereas, for the Cht-*g*-PVP-3 copolymer, it is higher than for the non-modified polysaccharide. This disparity can be attributed to the different distribution of grafted PVP chains within the copolymer macromolecules. The Cht-*g*-PVP-1 copolymer is characterized by a lower molecular weight of grafted PVP chains. Consequently, these chains almost do not contribute to the viscosity of chitosan. Conversely, the Cht-*g*-PVP-3 copolymer contains PVP chains at a higher molecular weight, which contributes to the viscosity. Additionally, it should be noted that the viscosity values indicate that we can minimize chitosan’s destruction during synthesis.

The results of the antibacterial activity of the copolymers obtained against *S. aureus* and *E. coli,* as well as an evaluation of their toxicity to eukaryotic cells, are shown in Table 3. As can be seen, both compounds demonstrated relatively low activity compared to the conventional antiseptic benzalkonium chloride. On the other hand, Cht-*g*-PVP can be used for the coating of food products and thus provide aseptic conditions. Of note, with the increase in the PVP content in the copolymers, their antibacterial activity decreases and their toxicity reduces significantly. Apparently, grafting PVP chains reduces the number of free amino groups, which are responsible for chitosan’s antibacterial activity. Thus, there is a decrease in their positive charge density and, consequently, with the growth of the PVP content in Cht-*g*-PVP copolymers, their antibacterial activity decreases. Also, the CC_50_ values show that the Cht-*g*-PVP copolymers are low-toxic, and this fact highlights the prospects for their application in the food industry.

## 4. Conclusions

In conducting this research, we optimized the synthesis conditions for Cht-*g*-PVP copolymers and achieved a 34.5% monomer conversion and 19.8% grafting efficiency in a grafting-to-chitosan reaction occurring in a green reaction medium without heavy metal ions or organic solvents. Grafting PVP chains alters the original chitosan’s structure and properties. The copolymers obtained are water-soluble due to the interruption of intra- and intermolecular hydrogen bonds and the presence of hydrophilic PVP chains. The surface of the copolymers is smoother compared to native chitosan. Also, the copolymers can form elastic films while preserving their antibacterial activity and viscosity properties, which makes them promising candidates for application in the food industry as gelling agents or in smart packaging development, which will be evaluated in our further research.

## Data Availability

The original contributions presented in the study are included in the article, further inquiries can be directed to the corresponding authors.

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
