# Peer review of "The Low-Waste Grafting Copolymerization Modification of Chitosan Is a Promising Approach to Obtaining Materials for Food Applications"

_polymers, 2024, doi:10.3390/polym16111596_

Round 1

Reviewer 1 Report

Comments and Suggestions for Authors

Dear Authors,

The work submitted to me for review titled "Low-Waste Grafting Copolymerization Modification of 2 Chitosan Is a Promising Approach to Obtaining Materials for 3 Food Applications" is part of the current research topic on innovative packaging materials by green chemistry requirements. The authors synthesized a new material in the copolymerization process of chitosan and PVP. They refined the reaction conditions to obtain a material with the best physicochemical parameters without using heavy metal ions or organic solvents. I have no objections to the substantive part, but I suggest the authors enrich it in accordance with the comments below. Since the authors propose using the obtained material to produce packaging, some information about the product's appearance, the physical form in which it was analyzed, and the possibilities of its shaping. It may be worth posting a photo of the final product. For the packaging industry, the best solution would be the ability to form films - have any attempts been made to obtain them? Furthermore, the analysis of the mechanical properties would be valuable. Figures 2 (a, b) require editorial correction. Uniform colors should be kept for the spectra of the same sample, but I suggest choosing dark colors; pale colors limit the readability of the drawings and increase the font - the descriptions are almost illegible.

Table 2: there is n(Cht)/n(VP) - should it be n(Cht)/n(PVP)?

Author Response

Dear Reviewer,

We would like to thank you for taking the time to read our manuscript. In view of your constructive criticism, we have revised the manuscript considerably.

In the following, we respond to particular concerns raised by you in a step-by-step manner. The changes in the text are marked in yellow.

Point 1. Since the authors propose using the obtained material to produce packaging, some information about the product's appearance, the physical form in which it was analyzed, and the possibilities of its shaping. It may be worth posting a photo of the final product. For the packaging industry, the best solution would be the ability to form films - have any attempts been made to obtain them? Furthermore, the analysis of the mechanical properties would be valuable.

Answer 1. Thank you for your valuable comment. Indeed, one of the possible uses of the obtained copolymers is the creation of smart food packaging. The samples used in this work were in the form of fine powders. In the future, we plan to produce films from these samples and test their mechanical properties. This will be the goal of our future work. We emphasized this in conclusion.

Point 2. Figures 2 (a, b) require editorial correction. Uniform colours should be kept for the spectra of the same sample, but I suggest choosing dark colours; pale colours limit the readability of the drawings and increase the font - the descriptions are almost illegible.

Answer 2. We edited Fig. 2 according you comment.

Point 3. Table 2: there is n(Cht)/n(VP) - should it be n(Cht)/n(PVP)?

Answer 3. No, n(Cht)/n(VP) is correct, since we studied the influence of chitosan/N-vinylpyrrolidone ratio on copolymer characteristics.

Thank you! You help us become better!

Reviewer 2 Report

Comments and Suggestions for Authors

In this manuscript, the authors present a promising study on the grafting copolymerization of chitosan for food applications. With some major revisions, the manuscript has the potential to make a significant contribution to the field.

Introduction:

1)     The authors report in the abstract that chitosan is employed in biomedicine. However, some references to this application should be included. For example:

·       ACS Appl. Eng. Mater. 2023, 1, 508−518

·       Int. J. Mol. Sci. 2020, 21(2), 487

·       ACS Appl. Bio Mater. 2020, 3, 11, 8075–8083

Material and methods:

1)     The authors should indicate the number of scans in the FTIR-ATR section. Moreover, they should change FTIR with FTIR-ATR throughout the manuscript.

2)     The authors should report how the porosity has been evaluated, since on page 10, lines 340-341, they state: “Additionally, a significant number of pores varying in diameter, shape, and depth up to 40 nm are observed. The maximum height difference is 94.3 nm, and the average roughness is 8.79 nm.

Results and discussion:

1)                 In Figure 2, please change the spectra colours since it is difficult to appreciate the bands. Moreover, the spectra should be stacked for the grafting evaluation.

2)                 For further information, the authors should also analyse the vinylpyrrolidone monomer by FT-IR ATR and 1H-NMR.

3)                 Regarding the analysis of copolymers by 1H-NMR, the authors could integrate signals to confirm that the narrow peaks are related to grafted poly(N-vinylpyrrolidone) chains. Otherwise, the products should be purified further, for example, by dialysis and semipreparative GPC.

4)                 Please uniform the digits for the NMR chemical shift. The authors on page 8, lines 314-317, state: “The NMR spectra of Cht-g-PVP copolymers contain both the aforementioned and new signals. The signals of PVP backbone protons are located in the regions of 1.2–1.53 and 3.4–3.56 ppm, while the lactam ring protons are reflected in the regions 2.11–2.14, 2.2–2.5, and 3.41–3.43 317 ppm [32] (Fig. 3B, C).

5)                 In figure 4, the authors should indicate the scale for a, b, c, and d. Moreover, they should correct caption d.

Author Response

Dear Reviewer,

We would like to thank you for taking the time to read our manuscript. In view of your constructive criticism, we have revised the manuscript considerably.

In the following, we respond to particular concerns raised by you in a step-by-step manner. The changes in the text are marked in yellow.

Point 1. Introduction:

The authors report in the abstract that chitosan is employed in biomedicine. However, some references to this application should be included. For example:

ACS Appl. Eng. Mater. 2023, 1, 508−518

Int. J. Mol. Sci. 2020, 21(2), 487

ACS Appl. Bio Mater. 2020, 3, 11, 8075–8083

Answer 1. Recommended references were added to the MS.

Point 2. Material and methods:

1) The authors should indicate the number of scans in the FTIR-ATR section. Moreover, they should change FTIR with FTIR-ATR throughout the manuscript.

2) The authors should report how the porosity has been evaluated, since on page 10, lines 340-341, they state: “Additionally, a significant number of pores varying in diameter, shape, and depth up to 40 nm are observed. The maximum height difference is 94.3 nm, and the average roughness is 8.79 nm.

Answer 2. Required information was added to the Materials and Method part.

Results and discussion:

Point 1. In Figure 2, please change the spectra colours since it is difficult to appreciate the bands. Moreover, the spectra should be stacked for the grafting evaluation.

Answer 1. Fig. 2 was edited according to your recommendation.

Point 2. For further information, the authors should also analyse the vinylpyrrolidone monomer by FT-IR ATR and 1H-NMR.

Answer 2. Thank you for your valuable comment. The FTIR and 1H NMR spectra of N-vinylpyrrolidone are known and widely published data. These results are published, for example, here: https://www.sciencedirect.com/science/article/pii/S2352492823004002

or https://www.mdpi.com/1996-1944/11/12/2535

or https://link.springer.com/article/10.1007/s00396-022-04979-x .

Moreover, they do not contradict the results obtained in our studies.

Point 3. Thank you for your valuable comment. Regarding the analysis of copolymers by 1H-NMR, the authors could integrate signals to confirm that the narrow peaks are related to grafted poly(N-vinylpyrrolidone) chains. Otherwise, the products should be purified further, for example, by dialysis and semipreparative GPC.

Answer 3. Of course, such different signal widths of PVP and chitosan in the NMR spectrum may suggest that there are two separate components in the system. Therefore, we dialyzed the copolymers in cellophane bags, cut off 14 kDa, against distilled water. The spectra of the purified polymers did not show significant changes in the NMR spectra. Therefore, the difference in signal width is due to the influence of the molecular weights of the products.

The copolymer compositions were determined from the results of their acid destruction (Eq. 4 in the Materials and Methods part).

Point 4. Please uniform the digits for the NMR chemical shift. The authors on page 8, lines 314-317, state: “The NMR spectra of Cht-g-PVP copolymers contain both the aforementioned and new signals. The signals of PVP backbone protons are located in the regions of 1.2–1.53 and 3.4–3.56 ppm, while the lactam ring protons are reflected in the regions 2.11–2.14, 2.2–2.5, and 3.41–3.43 ppm [32] (Fig. 3B, C).

Answer 4. Data and Fig. 3 were corrected according to your recommendation

Point 5. In figure 4, the authors should indicate the scale for a, b, c, and d. Moreover, they should correct caption d.

Answer 5. Fig. 4 was edited according to your recommendation.

Thank you! You help us become better!

Round 2

Reviewer 1 Report

Comments and Suggestions for Authors

Dear Authors,

I recommend your valuable manuscript for publication in Polymers.

All the best.

Reviewer 2 Report

Comments and Suggestions for Authors

The manuscript has been accordingly improved and is suitable for publication.